# Retinal Gene Expression of Selective Genes and Histological Stages of Embryonic and Post-Hatch Chickens (*Gallus gallus*)

**DOI:** 10.3390/genes13112048

**Published:** 2022-11-06

**Authors:** Nasmah K. Bastaki, Vanessa R. Lobo, Thecla Gomes, Taybha A. Albarjes

**Affiliations:** 1Department of Biological Science, Faculty of Science, Kuwait University, Kuwait City 13060, Kuwait; 2Department of Bioinformatics and System Biology, University of Manchester, Manchester M13 9PL, UK

**Keywords:** retina, gene expression, histology staining, embryonic chick, adult chicken

## Abstract

Chickens are excellent models for the study of retinal development and function. Gene expression at the correct time is crucial to retinal development and function. The present study aimed to investigate retinal gene expression and morphology in locally grown chickens at various developmental stages. RNA was extracted from the retina at the embryonic and post-hatch stages, and the retinal layers were stained with haematoxylin and eosin (H&E). RT-PCR and RT-qPCR were used for gene expression analysis of 14 selected genes. The results showed that all the retinal genes were expressed at different developmental stages. However, there were slight noticeable variations in expression patterns. At the morphological level, all retinal layers were well observed, except for the outer plexiform layer that became visible in the fifteen-day chick embryo. The current study provides a baseline for standard retinal gene expression of 14 genes and retinal histological staining. The selected genes have different roles in retinal development and function, and most of these genes are associated with retinal diseases. The results obtained here can be applied to molecular retinal research and retinal diseases with genetic factors in retina animal models or human diseases.

## 1. Introduction

Most vertebrates rely heavily on vision as their main sensory system. This dependence on visual sensory modality has led to a dramatic specialization of the retinal region of the eye [1]. The retina, an extension of the central nervous system (CNS), is a light-sensitive tissue that arises from the ectoderm during embryonic development. Its main function is relaying visual information to the brain so that it can be processed and interpreted [2].

The retina consists of a variety of neurons that are precisely organized within defined layers. Regarding the neuronal cells located within these layers, it is known that ganglion cells are located within the ganglion cell layer (GCL), amacrine cells are located within the inner plexiform layer (IPL), bipolar cells are located within the inner nuclear layer (INL), horizontal cells are located within the outer plexiform layer (OPL), and rods and cones are located within the outer nuclear layer (ONL). No neuronal cells exist in the nerve fiber layer (NFL) or retinal pigment epithelium (RPE) [3].

Chicken embryos are a classic model system for studying eye development and retinal function [4]. The chick has a cone-rich retina, with cellular responses similar to those of humans [5]. The chicken *Gallus domesticus*, has been used in vision research for over a century [5]. Chicken eyes at the embryonic and adult stages are easy to manipulate because of their large size, and their anatomy resembles that of humans. Chickens are also easy to handle and maintain. Moreover, embryogenesis in chickens occurs externally, enabling easy access to the embryonic tissues and organs. The genome of chickens has been completely sequenced, and many genes corresponding to different eye diseases in humans have homologs in chickens [6]. All these advantages make chicken an excellent model for ocular genetics research and ocular diseases.

At the molecular level, a different set of genes controls the retina’s ability to function correctly. Some genes are involved in the early development of the retina during embryogenesis, while others are involved at later stages of the development—during and after maturation [7]. The identification of what is known as “disease genes” or “mutated genes” in the retina can dramatically affect its function, resulting in deficient eyesight and major impairment of vision [2].

Previous studies have identified candidate genes that might be involved in refractive development. In these studies, induced chicken models of myopia and ametropia were used for retinal transcriptome analyses. The retinas of myopic and ametropic chickens were used in gene microarrays to screen for candidate genes with altered retinal gene expression [8,9,10]. In these studies, some candidate genes were validated by reverse-transcription quantitative PCR (RT-qPCR), whereas others were not validated. There is a need to validate the expression patterns of candidate retinal genes that are involved in normal retinal function and development throughout the developmental stages of the retina. Moreover, the expression patterns of most candidate genes were not confirmed at the embryonic stage of retinal development.

The objective of the present study was to select 14 genes, based on previously reported studies on retinal transcriptome analysis in chickens [9,10,11], and to assess their retinal gene expression patterns using locally grown chickens at different developmental stages. Table 1 summarizes the full names and abbreviations of the genes used in the present study as well as their gene expression, as confirmed by RT-qPCR, in the adult retina and not embryonically.

In this study, two embryonic stages (eight-day chick embryos (E8) and fifteen-day chick embryos (E15)), two post-hatch stages (new-hatch chicks (P2), and adult chickens (P50)) were used to assess retinal gene expression patterns throughout the different developmental stages. Moreover, morphological differentiation of the developing retina was conducted with histological assessment of the retinal layers at all developmental stages.

## 2. Materials and Methods

### 2.1. Ethical Statement

This research was conducted at Kuwait University, Kuwait. The chickens were cared for under the supervision of a poultry veterinarian at Kuwait University. They were handled, maintained, and experimented with according to the *Instructions Guide for the Care and Use of Laboratory Animals* [14]. The experimental procedures were approved by the Ethics Committee on Animal Use at Kuwait University ECULA: Ethical Committee for the Use of Laboratory Animals, DBS/IRB (ECULA)20-005.

### 2.2. Animal Source for the Study

A total of 120 fertilized eggs were obtained from local farms in Kuwait (Al-Wafra and Al-Abdali farms). Fertilized eggs were kept in an egg incubator (MG 100/150, Film, Italy). Optimal growth conditions were maintained (temperature of 37.5 °C and humidity of 60%) until the embryos reached the desired time in development: eight-day (E8) and fifteen-day (E15) chick embryos. Newly hatched chicks (P2) from the incubator were raised and some were sacrificed after one week, while others were sacrificed after two months and used for the last stage of development (P50). Chickens were raised outdoors at a temperature of 35–38 °C and had full access to chicken food and water at all times. At the desired stage of development, chicks and chickens were euthanized, their eyes were enucleated, and the retinas were collected and immediately processed for histology and RNA extraction.

### 2.3. Histology

Chick eyeballs from the appropriate developmental stages were removed. Forty samples were used for the histology (10 samples per stage). The eyeballs were pricked with a needle so that the fixative could reach the inside. The cells were then fixed in 4% paraformaldehyde (P6148-500G, Sigma-Aldrich, St. Louis, MO, USA) and prepared with 2% sucrose in 0.1% M phosphate buffer (pH 7.4 needed to be maintained) for 24 h at 4 °C in the refrigerator. The next day, they were washed in PBS three times and dehydrated in an increasing percentage of alcohol starting from 70%, 80%, and 90% for an hour each, to 100% for 3 h with two changes. The eyeballs were then cleared in xylene (131769, Panreac, Barcelona, Spain) for 2 h and filtered through paraffin wax (P3558, Sigma-Aldrich). A tissue processor (Citadel 2000, Shandon, CA, USA) was used for the above processing. The filtered tissues were embedded in paraffin blocks using a Histo-Embedder (38621438, Leica, Wetzlar, Germany). The blocks were cut, and vertical sections were made at 4–5 μm thickness using a Rotary Microtome (Leica RM2235), stained with Meyer’s hematoxylin (MHS16, Sigma-Aldrich) and eosin (102439, Sigma-Aldrich), and mounted with DPX (10197905000, Merck, Darmstadt, Germany).

### 2.4. RNA Extraction and cDNA Synthesis

Thirty-two samples were used for the RNA extraction (eight samples per stage). RNA was extracted from the retinas according to the TRIzol protocol (15596018, Invitrogen, Thermo Fisher Scientific, Waltham, MA, USA) and resuspended in 50 µL DEPC-treated RNAse-free water. The quality and integrity of the RNA samples were checked using a Nanodrop 8000 Spectrophotometer (ND-8000-GL, Thermo Fisher Scientific). Genomic DNA was removed using a TURBO DNA-free kit (AM1907, Invitrogen). cDNA was synthesized using a high-capacity cDNA reverse transcription kit and oligo dT primers, according to the manufacturer’s protocol (4368814, Applied Biosystems, Thermo Fisher Scientific, Lithuania). The quality and integrity of the cDNA were checked using a Nanodrop 8000 Spectrophotometer.

### 2.5. RT-PCR

Based on previously published microarray data on chicken retinas, 14 retinal genes were chosen [9,10,11]. Table 2 shows the 14 genes and their mRNA codes, as listed on PubMed. Forward and reverse primers were designed for the 14 genes using the Primer-BLAST primer design tool at NCBI (http://www.ncbi.nlm.nih.gov/tools/primer-blast/ (accessed on 30 June 2017)) to amplify products of approximately 150–200 base pairs (bp) (Table 2). For PCR, Platinum^TM^ Green Hot Start PCR 2X master mix (13001014, Invitrogen) was used according to the manufacturer’s protocol. The concentration of cDNA for all developmental stages was maintained at 25 ng for all PCR reactions. The PCR conditions were as follows: initial denaturation at 94 °C for 2 min, cycle denaturation: 94 °C for 30 s, annealing temperature: 55 °C for 30 s, extension temperature was 72 °C for 30 s. The PCR cycle was repeated 35 times with a final extension of 5 min. For gel electrophoresis, 12 µL of each PCR reaction was loaded onto a 1.5% agarose gel and stained with SYBR Safe DNA Gel Stain (S33102, Invitrogen).

### 2.6. Sanger Sequencing 

PCR bands were purified using the Purelink Quick Gel Extraction and PCR Purification Combo Kit (K220001, Invitrogen). The products were prepared for sequencing using a BigDye Terminator v3.1 Cycle Sequencing Kit (4337455, Applied Biosystems, Waltham, MA, USA) and a BigDye XTerminator Purification Kit (4376486, Applied Biosystems). Sanger sequencing was performed at the Kuwait University Biotechnology Center Sequencing Facility (ABI 3130xI Genetic Analyzer, Thermo Fisher Scientific, Japan). The results of the sequencing were compared to retinal genes from the online available NCBI databases, confirming the identity of all genes in the study.

### 2.7. RT-qPCR

RT-qPCR was performed on a Bio-Rad CFX96 Real-Time System (C1000 Touch Thermal Cycle, Bio-Rad, Singapore). RT-qPCR was conducted on cDNAs from E8, P2, and P50; E15 was not included as the sample number for this stage was not sufficient to proceed with RT-qPCR. For every developmental stage, the sample number used for the RT-qPCR was six, collected from both eyes of three embryos in E8, three chicks in P2, and three chickens in P50. Each cDNA sample was tested in duplicate using the PowerUp SYBR Green Master Mix 2X (A25779, Applied Biosystems) and gene-specific primers (Table 2). Reactions were performed in duplicate in a total of 10 µL reaction mixture containing 5 µL PowerUp SYBR Green Master Mix, 2 µL of diluted cDNA, 1 µL of each specific primer (10 µM), and 1 µL of molecular-grade water. The PCR conditions were as follows: 50 °C for 2 min, (40 cycles of 95 °C for 2 min, 95 °C for 15 s, 57 °C for 15 s, 72 °C for 1 min), and an extra step of melting analysis for each sample at temperature 65 °C for 5 s and 95 °C for 5 s, as well. RT-qPCR results were calculated using the ΔCt value (Ct target gene- Ct GAPDH). Statistical analysis was performed using the Student’s *t*-test using the Bio-Rad CFX96 Real-Time System to obtain the *p*-value and the standard error of the mean. Statistical significance was set at *p* < 0.05.

## 3. Results

### 3.1. Morphological Differentiation Stages of the Developing Retinal Layers

#### 3.1.1. Eight-Day and Fifteen-Day Chick Embryos

Eyeballs from eight-day (Figure 1a,b) and fifteen-day chick embryos (Figure 2a,b) were histologically processed to observe the morphology of the retinal layers. Histological staining of retinal sections of the eight-day chick embryos using H&E stain showed the following layers (Figure 1c): NFL, GCL, IPL, INL, ONL, and RPE. In fifteen-day chick embryos, all the previously mentioned layers were well observed (Figure 2c), and the OPL became visible between the INL and the ONL.

#### 3.1.2. Newly Hatched Chick and the Adult Chicken

Eyeballs from newly hatched chicks (Figure 3a,b) and mature adult chickens (Figure 4a,b) were histologically processed to observe the morphology of the retinal layers. All the previously mentioned layers were visible and easily identified in the sections of the newly hatched chicks (Figure 3c) and the adult chicken (Figure 4c). The layers were ordered as before: the NFL, GCL, the IPL, INL, OPL, ONL, and RPE.

### 3.2. Gene Expression Analysis Using RT-PCR

Our RT-PCR results showed that all 14 genes (Table 2) were expressed and amplified as expected in the four developmental stages: eight-day chick embryo, fifteen-day chick embryo, newly hatched chick, and adult chicken (Figure 1d, Figure 2d, Figure 3d and Figure 4d). However, the intensities of the bands for each gene were different, even though the same cDNA concentrations were used for each gene.

In three of the genes, an extra band was observed in some developmental stages. For example, for *PHLDA2*, an upper band of 1500 bp was observed in the fifteen-day chick embryo (Figure 2d, band #15) and adult chicken (Figure 4d, band #14), in addition to the expected band of 144 bp, which was not visible in the retinas of eight-day chick embryos (Figure 1d, band #13) and newly hatch chicks (Figure 3d, band #14). When the 1500 bp band was sequenced and aligned against the NCBI database by BLAST, it was an exact match to *Gallus* pleckstrin homology like domain family A member 2 (PHLDA2). Similar patterns were observed for Dual Specificity Phosphatase 4 (*DUSP4*); in all developmental stages, there was an additional band of 250 bp in addition to the 147 bp band (Figure 1d, band #3; Figure 2d, band #4; and Figure 4d, band #3), except for the newly hatched chick (Figure 3d, band #3). When the 250 bp band was processed for sequencing and aligned against the NCBI database by BLAST, it was an exact match to *Gallus DUSP4*. The third gene that showed an additional band in RT-PCR was Myosin Heavy Chain 13 (*MYH13*). In addition to the expected 149 bp band, an upper band with a size of 300 bp was observed at all stages, which was most intense in the mature adult chicken (Figure 4d, band #9). When the 300 bp band was processed for sequencing and aligned against the NCBI database by BLAST, it was an exact match to *Gallus* MYH13 (data for sequencing are not shown).

### 3.3. Gene Expression Analysis Using RT-qPCR

For RT-qPCR, three developmental stages were tested: eight-day chick embryo (E8), newly hatched chick (P2), and adult chicken (P50) (Figure 5). The fifteen-day chick embryo (E15) was not included in the RT-qPCR as the number of samples for that particular stage was not sufficient to proceed with, and, therefore, was eliminated from this part.

The expression patterns of the 13 genes in RT-qPCR showed that all genes were expressed in E8, P2, and P50. There were slight noticeable variations in the relative quantity of mRNA, and some genes showed similar expression patterns. For example, some genes were expressed slightly more in E8 and P2 than in P50, such as *MYH13*, *CTGF*, and *NOG*, while others were expressed slightly more in E8 than in P2 and P50, such as *VIP* and *HPRT1*. A few genes showed similar expression patterns during all the developmental stages, such as *BMP2*, *DUSP4*, and *ELF1*. Other genes were expressed more in E8 and P50, but their expression decreased in the middle stage of P2, such as *OSBPL6* and *GHRHR*. This was extremely noticeable in *NTS*, as its expression decreased by more than a three-fold difference in P2 compared with E8 and P50. In comparison with *OSBPL6*, *GHRHR*, and *NTS*, where their expression decreased in P2 only, *PHLDA2* showed the opposite results, as its expression was higher in P2 than in E8 and P50. In contrast, *GRB2* expression was slightly higher in P50 than in E8 and P2.

## 4. Discussion

The present study describes the age-dependent morphological differentiation stages of developing retinal layers of chick embryos and adult chickens locally grown in Kuwait. It also describes the gene expression patterns of 14 selected genes based on previously published chicken microarray data conducted on induced chicken models of refractive errors [9,10,11].

The retinae of chicken embryos (E8 and E15), newly hatched chicks (P2), and adult chickens (P50) were studied histologically. Our observations of the morphological stages of the retinal layers using H&E staining were consistent with those of previous studies [3,15]. All retinal layers were clearly visible and ordered as follows: the NFL, GCL, IPL, INL, OPL, ONL, and RPE. These retinal layers appeared clearly in all developmental stages except the OPL, which was not observed in eight-day chick embryos but was observed at the other developmental stages. This is consistent with the results of previous studies [15].

RT-PCR was used to assess gene expression in the retina at different developmental stages. All 14 genes selected in this study were expressed at E8, E15, P2, and P50. RT-qPCR was used to quantify the relative amounts of mRNA in E8, P2, and P50. GAPDH was used as a housekeeping gene and a gene normalizer. To the best of our knowledge, this is the first study in the Arabian Gulf region to investigate retinal gene expression in locally grown chickens across different developmental stages.

Our results not only validated the gene expression of candidate retinal genes previously reported in microarray studies but are also the first to report the gene expression patterns of a few genes in the retina, such as *NTS*, *MYH13*, and *PHLDA2*, at the embryonic and post-hatch stages. Moreover, to the best of our knowledge, most of the candidate genes selected in this study have not been validated before at the embryonic stages of the retina using RT-qPCR in most organisms, especially embryonic chickens.

The 14 genes used in this study have different functions in the retina and eye. Table 3 lists their known functions based on literature reviews. These genes have potential uses as therapeutic agents for different diseases of the retina and eye (Table 3).

The results obtained herein can be applied to molecular retinal research and retinal disorders. Future studies can be performed using chicken models of common disorders, such as diabetic retinopathy, macular degeneration, and retinal vein occlusion. Diabetic retinopathy, a microvascular disease of the retina, is a major complication of diabetes mellitus that leads to blindness. Previous studies have identified several candidate genes associated with the progression of diabetic retinopathy; examples of such candidate genes that were selected in our studies and found to be associated with diabetic retinopathy are *GRB2* [33,40], *GHRHR* [39], *BMP2* [19,41], and *GAPDH* [20,42]. Future studies should target these genes in chicken models of diabetic retinopathy.

In conclusion, this study provides a baseline for standard retinal expression of 14 genes and retinal histological staining of locally grown chickens. All 14 retinal genes selected in this study were expressed in the embryonic and adult stages. The relative quantity of retinal gene mRNA showed slight variations in gene expression during the embryonic and post-hatch stages. However, a few genes showed similar expression patterns. At the morphological level, retinal layers were consistently observed, as reported in other studies.

## Figures and Tables

**Figure 1 genes-13-02048-f001:**
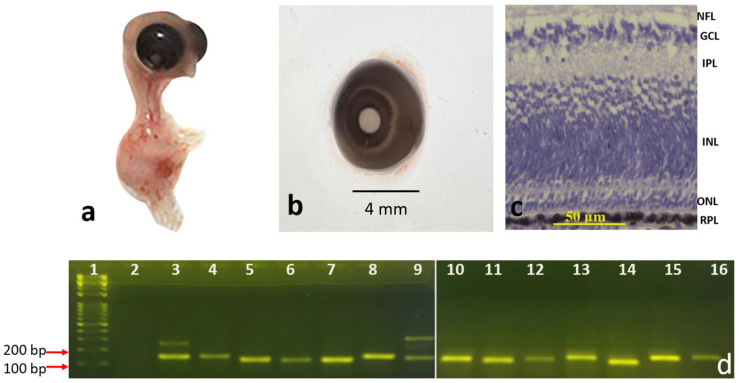
Eight-day chick embryos: (**a**) whole embryo, (**b**) the retrieved eye, (**c**) cross section of the retina (20X); scale bar = 50 μm (**d**) RT-PCR of the different genes labeled as follows: 1: 100 bp DNA marker, 2: negative control, 3: *DUSP4*, 4: *ELF*, 5: *GAPDH*, 6: *GHRHR*, 7: *GRB2*, 8: *HPRT*, 9: *MYH13*, 10: *NOG*, 11: *NTS*, 12: *OSPBL6*, 13: *PHLDA*, 14: *VIP*, 15: *BMP2*, 16: *CTFG*.

**Figure 2 genes-13-02048-f002:**
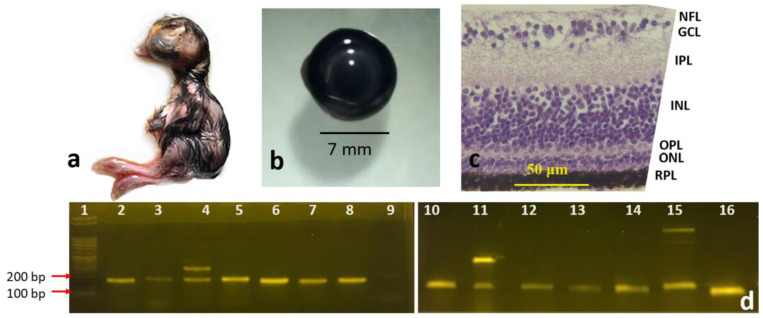
Fifteen-day chick embryos: (**a**) whole Embryo, (**b**) the retrieved eye, (**c**) cross section of the retina (20X); scale bar = 50 μm (**d**) RT-PCR of the different genes labeled as follows: 1: 100 bp DNA marker, 2: *BMP2*, 3: *CTGF*, 4: *DUSP4*, 5: *ELF,* 6: *GAPDH*, 7: *GHRHR*, 8: *GRB2*, 9: negative control, 10: *HPRT*, 11: *MYH13*, 12: *NOG*, 13: *NTS*, 14: *OSBPL*, 15: *PHLDA*, 16: *VIP*.

**Figure 3 genes-13-02048-f003:**
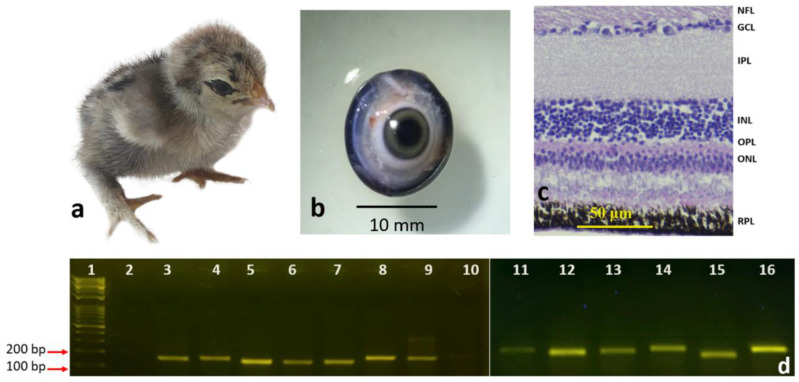
Newly hatched chick: (**a**) whole chick, (**b**) the retrieved eye, (**c**) cross section of the retina (20X); scale bar = 50 μm (**d**) RT-PCR of the different genes labeled as follows: 1: 100 bp DNA marker, 2: negative control, 3: *DUSP4,* 4: *ELF1,* 5: *GAPDH,* 6: *GHRHR,* 7: *GRB2,* 8: *HPRT,* 9: *MYH13,* 10: *CTGF,* 11: *NOG,* 12: *NTS,* 13: *OSBPL,* 14: *PHLDA,* 15: *VIP,* 16: *BMP2*.

**Figure 4 genes-13-02048-f004:**
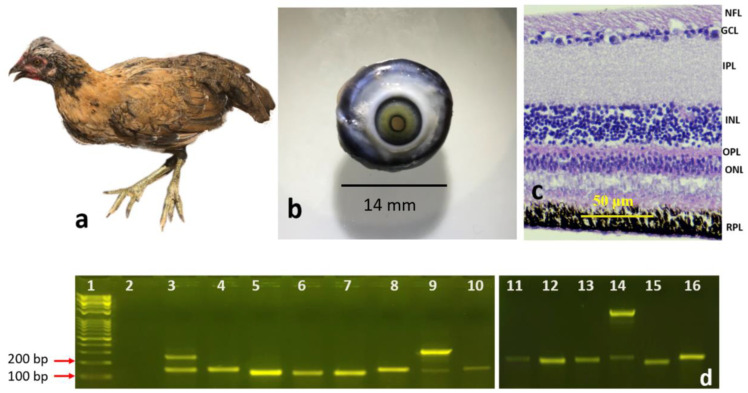
The adult chicken: (**a**) whole chicken, (**b**) the retrieved eye, (**c**) cross section of the retina (20X); scale bars = 50 μm (**d**) RT-PCR of the different genes labeled as follows: 1: DNA marker, 2: negative control, 3: *DUSP4,* 4: *ELF2,* 5: *GAPDH,* 6: *GHRHR*, 7: *GRB2,* 8: *HPRT,* 9: *MYH13,* 10: *CTGF,* 11: *NOG,* 12: *NTS,* 13: *OSPBL,* 14: *PHLDA,* 15: *VIP,* 16: *BMP2*.

**Figure 5 genes-13-02048-f005:**
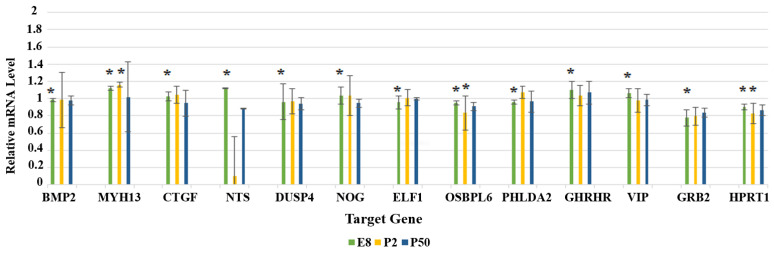
The relative mRNA expression levels of the 13 studied genes in the retina of an eight-day chick embryo (E8), newly hatched chick (P2), and adult chicken (P50) as detected by RT-qPCR. Results are presented as the mean (±SEM) mRNA expression (n = 3 in each group, * *p* < 0.05).

**Table 1 genes-13-02048-t001:** List of retinal genes used in the present study (full and abbreviated names) and whether their expression has been validated by RT-qPCR in previous studies (in adult-stage retinas). The previous studies are listed in parentheses.

*Gallus* Gene Full Name	Gene Abbreviation	Validation with RT-qPCR [Reference]
*BMP2*	Bone Morphogenetic Protein 2	Yes [8,10]
*GAPDH*	Glyceraldehyde-3-Phosphate Dehydrogenase	Yes [12,13]
*NTS*	Neurotensin	Not confirmed before this study
*PHLDA2*	Pleckstrin Homology Like Domain family A member 2	Not confirmed before this study
*CTGF*	Connective Tissue Growth Factor	Yes [8,13]
*DUSP4*	Dual Specificity Phosphatase 4	Yes [10]
*VIP*	Vasoactive Intestinal Peptide	Yes [8,10]
*NOG*	Noggin	Yes [10]
*OSBPL6*	OxySterol Binding Protein-Like 6	Yes [9,10]
*MYH13*	MYosin, Heavy chain 13, skeletal muscle	Not confirmed before this study
*GRB2*	Growth factor Receptor Bound protein 2	Yes [9]
*HPRT1*	Hypoxanthine Phospho-Ribosyl Transferase 1	Yes [12]
*ELF1*	E74 Like ETS transcription Factor 1	Yes [9]
*GHRHR*	Growth Hormone Releasing Hormone Receptor	Yes [9]

**Table 2 genes-13-02048-t002:** List of NCBI codes (mRNA) for the 14 genes used in the current study, their primers, and their expected PCR amplified sizes in base pairs (bp).

Gene Abbreviation	NCBI Code (mRNA)	Primers Sequence (5′–3′) F: Forward R: Reverse	PCR Fragment Length (bp)
*BMP2*	NM_204358.1	F: GCCAGAAACAAGTGGGAAAA R: TACGGTGATGGTAGCTGCTG	149 bp
*GAPDH*	NM_204305.1	F: CACACAGAAGACGGTGGATG R: CAGCTCAGGGATGACTTTCC	124 bp
*NTS*	NM_001277360.1	F: AAGACAGTTCCCTGCTGCTC R: GATCAAATGCGTCTTGCTGA	128 bp
*PHLDA2*	NM_001199595.1	F: GCCATCCAGGACTTCAAGAG R: CCTGTCCCTTTTCAGTCCAA	144 bp
*CTGF*	NM_204274.1	F: TTCCAGAGCAGCTGCAAGTA R: ACCCACTCCTCACAGCACTT	149 bp
*DUSP4*	NM_204838.1	F: TGCCTCAGAGTATCCCGAAT R: AAGGGAAGGATTTCCACAGG	147 bp
*VIP*	NM_205366.2	F: AAATCCAGCCAAACTTCGAG R: GGTGTCCTTCAGAGGTCCAA	119 bp
*NOG*	NM_204123.1	F: CTCGGGGTAGACGATCTGG R: CAGCTTCTTGCTCAGCCTGT	138 bp
*OSBPL6*	XM_421982.5	F: CGAAGATGAGTTCGGAGGAG R: AGCCTTCCGTTCCAAAAGAT	134 bp
*MYH13*	XM_015295194.	F: CTGCGGATATCGAAACCTGT R: TAGGGGTTGGTCGAGATGAG	148 bp
*GRB2*	NM_204411.1	F: ATGTGCAGCAGTTCAAGGTG R: GCTGGTTCCTGGAGACAGAT	123 bp
*HPRT1*	NM_204848.1	F: AGCCCCATCGTCATATGCT R: AGCCCCATCGTCATATGCT	155 bp
*ELF1*	NM_001006269.1	F: GGAAATGCCAAAAGATCTCG R: TCCTCCCTTTGTTCCTGTGT	155 bp
*GHRHR*	NM_001037834.1	F: CTACGCTGCCCCAGAAATTA R: AAAGCTGCAATGGTCAATGTC	123 bp

**Table 3 genes-13-02048-t003:** List of the known function of the 14 retinal genes used in this study and their association with diseases or potential use in therapy (based on literature reviews).

*Gallus* Gene	Function in the Retina or Eye	Disease-Associated-Therapeutic
*BMP2*	Retinal development Retinal growth regulation Embryogenesis [16]	Myopia and hyperopia [16,17] Diabetic retinopathy [18,19] Association with ametropia [10]
*GAPDH*	Housekeeping proteins (glycolysis) [20]	Diabetic retinopathy [20]
*TS*	Retinal neural development [21] Anti-apoptotic role [22]	Association with myopia [8]
*PHLDA2*	Not listed in the literature	Association with myopia [8]
*CTGF*	Vessel growth during early retinal development [23]	Ischemic retinopathies [23] Angiogenesis [23] Wound healing [24] Myopia [8]
*DUSP4*	Differentiation Proliferation Survival Apoptosis [25]	Hyperopia [10]
*VIP*	Neuromodulator [26] Stimulation of cell division and differentiation of RPE cells [8]	Myopia [26]
*NOG*	Early eye growth and development of retinal structures [27,28]	Therapeutic in neurodegenerative retinal diseases [29] Association with ametropia [10]
*OSBPL6*	Neural retina development [30]	Association with ametropia [10]
*MYH13*	Eye muscle movement and coordination [31]	Association with ametropia [10]
*GRB2*	Insulin pathway [9] Appropriate functioning of rods (photoreceptor cells) by -insulin pathway [32] Excessive eye growth [9]	Diabetic retinopathy [33] Myopia [9]
*HPRT1*	Housekeeping gene in the retina [34,35]	Neural development [36]
*ELF1*	Retinotectal projection [9] ElF1 and its receptors act as retinal exon guidance [37]	Myopia [9]
*GHRHR*	Protective roles in the eye [38] GHRH diminishes retinal neurovascular injury in the early stages of diabetic retinopathy [39]	Retinoblastoma [38] Antioxidant and anti-inflammatory [39] Myopia [9]

## Data Availability

The authors declare that all the data supporting the findings of this study are available within the article or from the corresponding author upon reasonable request.

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
