# Peer review of "Retinal Gene Expression of Selective Genes and Histological Stages of Embryonic and Post-Hatch Chickens (Gallus gallus)"

_genes, 2022, doi:10.3390/genes13112048_

Round 1

Reviewer 1 Report

The manuscript by Bastaki et al,, provides important baseline data for standard retinal expression of selected genes in the domestic chicken.

This is an interesting study, providing important introductory data for refractory development in the domestic chicken,

I only had one comment which I would like the authors to consider:

Discussion: Can the authors elaborate in further detail as to how their data provides the baseline for the standard retinal expression, and what would be the next steps/priorities for future study?

Reviewer 2 Report

The goal of this study was to document the retinal development in the domestic chicken, using histology, RT-PCR and rt-qPCR. Four different stages were compared: E8, E15, P2 and P50. A set of 14 genes was analyzed; some of these have not yet been analyzed in this context. 

General comments: 

The manuscript is in general well written. The authors mention that they used in total 120 fertilized eggs, but they should have stated how many samples were used for each experiment. How many times were the PT-PCR and rt-qPCR experiments repeated? Immunohistochemistry in addition to H-E staining would have increased the value of the study. 

Specific comments:

p. 3, line 10 of section 2.2: “eyes pulled out” – should be “eyes enucleated”

The histology panels in Figures 1-4 need scale bars. 

Figure 5: There are no error bars for the relative mRNA levels in Figure 5.

Reviewer 3 Report

This manuscript shows the morphological structure of the retina of local native chickens at 4 different developmental stages, and gene expression analysis of 14 selected genes associated with retinal diseases at 3 different developmental stages was performed by RT-PCR and RT-qPCR, but RT-qPCR lacked statistical significance tests.

The following points should be addressed:

(1)The figures of the morphology of the eye at the 4 different developmental stages should have a uniform scale to give the readers an idea of the true size of the eye at that stage. The figures of H&E should also be added to the scale.

(2)The primers lacked specificity.

(3)The RT-qPCR lacks a statistical significance test, and the results are not of extensive value, which cannot be used as strong evidence that the expression of these genes changes significantly at different developmental stages.

(4)The language expression needs to be optimized.

Round 2

Reviewer 2 Report

The authors have addressed the reviewers' comments, and significantly improved their manuscript and the figures.